# Clinical Validation of a Machine Learning-Based Biomarker Signature to Predict Response to Cytotoxic Chemotherapy Alone or Combined with Targeted Therapy in Metastatic Colorectal Cancer Patients: A Study Protocol and Review

**DOI:** 10.3390/life15020320

**Published:** 2025-02-19

**Authors:** Duilio Pagano, Vincenza Barresi, Alessandro Tropea, Antonio Galvano, Viviana Bazan, Adele Caldarella, Cristina Sani, Gianpaolo Pompeo, Valentina Russo, Rosa Liotta, Chiara Scuderi, Simona Mercorillo, Floriana Barbera, Noemi Di Lorenzo, Agita Jukna, Valentina Carradori, Monica Rizzo, Salvatore Gruttadauria, Marco Peluso

**Affiliations:** 1Department for the Treatment and Study of Abdominal Diseases and Abdominal Transplantation, Istituto di Ricovero e Cura a Carattere Scientifico-Istituto Mediterraneo per i Trapianti e Terapie ad Alta Specializzazione (IRCCS-ISMETT), University of Pittsburgh Medical Center (UPMC), 90127 Palermo, Italy; dpagano@ismett.edu (D.P.); atropea@ismett.edu (A.T.); rliotta@ismett.edu (R.L.); fbarbera@ismett.edu (F.B.); ndilorenzo@ismett.edu (N.D.L.); ajukna@ismett.edu (A.J.); mrizzo@ismett.edu (M.R.); 2Department of Biomedical and Biotechnological Sciences, University of Catania, 95123 Catania, Italy; vincenza.barresi@unict.it (V.B.); chiara.scuderi@studium.unict.it (C.S.); s.mercorillo98@gmail.com (S.M.); 3Department of Surgical, Oncological and Oral Sciences, University of Palermo, 90127 Palermo, Italy; antonio.galvano@unipa.it (A.G.); viviana.bazan@unipa.it (V.B.); 4Tuscany Cancer Registry, Clinical Epidemiology Unit, ISPRO-Study, Prevention and Oncology Network Institute, 50139 Florence, Italy; a.caldarella@ispro.toscana.it; 5Cancer Prevention Laboratory, ISPRO-Study, Prevention and Oncology Network Institute, 50139 Florence, Italy; cristiana.sani@unipa.it (C.S.); gianpaolo.pompeo@unipa.it (G.P.); 6Research and Development Branch, Cancer Prevention Laboratory, ISPRO-Study, Prevention and Oncology Network Institute, 50139 Florence, Italy; v.russo@ispro.toscana.it (V.R.); v.carradori@ispro.toscana.it (V.C.); m.peluso@ispro.toscana.it (M.P.); 7Department of General Surgery and Medical-Surgical Specialties, University of Catania, 95123 Catania, Italy

**Keywords:** artificial intelligence, chemotherapy, targeted therapy, responders, radiomics, biomarkers, algorithm, colorectal cancer metastasis

## Abstract

Metastatic colorectal cancer (mCRC) is a severe condition with high rates of illness and death. Current treatments are limited and not always effective because the cancer responds differently to drugs in different patients. This research aims to use artificial intelligence (AI) to improve treatment by predicting which therapies will work best for individual patients. By analyzing large sets of patient data and using machine learning, we hope to create a model that can identify which patients will respond to chemotherapy, either alone or combined with other targeted treatments. The study will involve dividing patients into training and validation sets to develop and test the models, avoiding overfitting. Various machine learning algorithms, like random survival forest and neural networks, will be integrated to develop a highly accurate and stable predictive model. The model’s performance will be evaluated using statistical measures such as sensitivity, specificity, and the area under the curve (AUC). The aim is to personalize treatments, improve patient outcomes, reduce healthcare costs, and make the treatment process more efficient. If successful, this research could significantly impact the medical community by providing a new tool for better managing and treating mCRC, leading to more personalized and effective cancer care. In addition, we examine the applicability of learning methods to biomarker discovery and therapy prediction by considering recent narrative publications.

## 1. Introduction

Personalized treatments for metastatic colorectal cancer (mCRC) are currently limited to a small number of drugs targeting specific molecular markers, such as anti-VEGF, anti-EGFR for RAS wild-type tumors, encorafenib for BRAF (V600E)-mutated tumors, programmed death-ligand 1/programmed cell death protein 1 (PDL-1/PD-1) inhibitors for mismatch repair-deficient (dMMR)/high-microsatellite-instability (MSI-H) tumors, and KRAS tyrosine kinase inhibitors for KRAS G12C mutated tumors [1]. Therefore, it is essential to identify mCRC patients who will respond to specific therapeutic regimens before treatment. Currently, there are no available clinical criteria for the early identification of treatment response prior to therapy initiation.

Few therapeutic options are available for resistant mCRC, which can be difficult to address with non-cross-resistant anticancer drugs in second and subsequent lines of treatment [2]. Adaptive resistance can occur at any time during treatment, leading to therapeutic failures and subsequently high mortality rates [3].

In the field of personalized therapy, artificial intelligence (AI) has gained interest for its problem-solving abilities, decision-making processes, and development of predictive models [4]. Recently, we examined the capability of learning methods to construct prognostic and predictive models using AI and a meta-analytic approach to predict chemotherapy response, alone or combined with targeted therapy, in patients with mCRC [5]. Our results demonstrated that molecular biomarker-based signatures can distinguish responders from non-responders, accurately identifying mCRC patients who respond to therapy and those who do not. This encourages the use of AI in developing personalized decision-making processes for chemotherapy treatment.

In this project, we will synthesize public archive datasets and conduct a retrospective epidemiological study to examine a panel of molecular biomarkers, such as chromosomal instability, mutational profile, and the whole transcriptome in tumor samples from mCRC patients. Machine learning technology will then be used to develop and validate a predictive model capable of forecasting the response to chemotherapy, alone or combined with targeted therapy, in mCRC patients, classifying them as responders or non-responders to improve personalized treatment decision-making.

The primary objective of the study is to evaluate the efficacy of the machine learning algorithm in predicting the response to chemotherapy alone or combined with targeted therapy in mCRC patients. The secondary objective is to assess the algorithm’s effectiveness in distinguishing between responsive and non-responsive mCRC patients. The response to therapy with the identified biomarker signature will be studied in (a) 2277 CRC patients using public datasets from The Cancer Genome Atlas (TCGA) and Gene Expression Omnibus (GEO) and (b) a retrospective study analyzing formalin-fixed paraffin-embedded (FFPE) tumor samples from a cohort of metastatic or recurrent CRC patients who underwent surgery at ISMETT’s Surgical Unit over the past 10 years.

The novelty and importance of the present study arise from the fact that this work addresses a relevant clinical need: predicting therapy response in mCRC patients using molecular data and AI networks.

Recently, we conducted a systematic review and meta-analysis to study the ability of AI technologies to build predictive models in mCRC patients [5]. In this work, we showed that all studies conducted in this field provided promising findings in predicting the response to therapy or toxic side effects. Also, we found that the overall weighted means of the area under the receiver operating characteristic curve were 0.90, 95% C.I. 0.80–0.95 and 0.83, 95% C.I. 0.74–0.89 in training and validation data sets, respectively, demonstrating a good performance in discriminating response vs. non-response mCRC patients. Therefore, we decided to add a dedicated literature review section to this research study protocol, including recent AI studies from 2022 to 2024, to provide a comprehensive overview of related fields.

## 2. Materials and Methods

### 2.1. Summary Description

Despite therapeutic advancements, outcomes for patients with metastatic colorectal cancer (mCRC) remain unsatisfactory and variable due to intrinsic or acquired resistance to treatments. Molecular biomarkers have recently been used as inputs for artificial intelligence models to identify signatures that can guide physicians in selecting individualized chemotherapy. Therefore, we will conduct a retrospective epidemiological study and perform a synthesis of public archival datasets to examine chromosomal instability, mutational profiles, and whole-transcriptome expression in formalin-fixed, paraffin-embedded (FFPE) tumor samples from patients with mCRC. Machine learning technology will be used to develop and validate a predictive model of chemotherapy response, alone or in combination with targeted therapy, to improve personalized treatment decision-making processes for patients with mCRC. In Figure 1, the general information of the study protocol and financing institution are reported.

Description and Distribution of Activities for Each Operational Unit:

Unit 1: Responsible for the epidemiological plan at the ISPRO Institute, Florence, Italy and will oversee clinical, laboratory, statistical, and bioinformatic activities. It will be involved in validating and developing the supervised learning algorithm for therapy response in mCRC patients based on molecular biomarker data and identifying the newly constructed molecular classifier. Unit 1 will also participate in validating overall transcription results obtained from Affymetrix transcriptome arrays by testing selected long non-coding RNAs via RT-PCR.

Unit 2: Responsible for collecting archived FFPE tumor samples from CRC patients who underwent surgery at the ISMETT Surgical Unit, Palermo, Italy over the past 10 years. Samples will include primary or metastatic lesions from metastatic or recurrent CRC patients who received standard chemotherapy with or without anti-EGFR or anti-VEGF agents. All samples will be obtained before therapy. Clinical and pathological diagnosis data, including mCRC status at diagnosis and therapy, will be collected according to defined standard protocols. Follow-up for a and mortality will be adequately considered for the entire project duration.

Unit 3: Responsible for extracting and purifying RNA and DNA from FFPE samples at the University of Palermo, Palermo, Italy. It will analyze mutational profiles of 50 CRC-related genes using platforms like Illumina MiSeq and IonS5 (Thermo Fisher, Waltham, MA, USA). Additionally, it will examine copy number variants for a series of relevant genes [6,7].

Unit 4: Will perform whole-transcriptome and molecular karyotype analysis at the University of Catania, Catania, Italy, to analyze lncRNA signatures using next-generation human transcriptomes (Affymetrix HTA2.0). Whole-genome DNA analysis will be conducted using high-resolution SNP genotyping arrays. Chromosomal instability will be examined using high-resolution whole-genome DNA copy number and SNP genotyping arrays to measure chromosomal aberration rates in tumors and metastases from CRC patients. The unit will utilize a CRC tumor classification based on the BCNA number to identify patient subgroups with different prognoses and therapy responses. The activities of each unit throughout the project are reported in Table 1.

### 2.2. Specific Objectives and Experimental Design

#### 2.2.1. Background and Unmet Needs

Personalized treatments for metastatic colorectal cancer (mCRC) are currently limited to a small number of molecularly targeted drugs [1]. There is an unmet need for the early identification of mCRC cases that will respond to specific treatment regimens, as there are no tools available in clinical practice to predict treatment response before therapy initiation.

#### 2.2.2. Primary Objective

The primary objective of this study is to evaluate the efficacy of a machine learning algorithm based on chromosomal instability, mutational status, and whole-transcriptome data in predicting the response to chemotherapy, either alone or combined with targeted therapy, in patients with mCRC. The secondary objective is to assess the algorithm’s ability to discriminate between responsive and non-responsive mCRC patients. The response to therapy considering the aforementioned signature will be studied in (a) 2,277 CRC patients using public datasets from The Cancer Genome Atlas (TCGA) and Gene Expression Omnibus (GEO) and (b) a retrospective study where we will collect formalin-fixed, paraffin-embedded (FFPE) tumor samples from a cohort of metastatic or recurrent CRC patients who underwent surgery at the ISMETT Surgical Unit over the past 10 years.

Various performance metrics, such as AUC, SE, SP, accuracy (ACC), positive predictive value (PPV), negative predictive value (NPV), and patient survival estimates based on hazard ratios (HRs), will be used to evaluate the predictive power of different algorithms to correctly classify response versus non-response 6 or more months after the last chemotherapy cycle.

#### 2.2.3. Secondary Objective

The secondary objective is to analyze various biomarkers, such as mutational profiles and chromosomal instability, in mCRC tumor samples to obtain molecular data used as input for machine learning models aimed at predicting therapy response and classifying patients as responders and non-responders. In 1990, Fearon and Vogelstein proposed a progressive model of colon cancer development, highlighting major genetic changes associated with progression stages, known as the “Vogelgram” [8]. Today, patients with colorectal metastases with KRAS or NRAS mutations are considered non-responders or poorly responsive to anti-EGFR therapy, while the presence of the BRAF V600E mutation identifies a subgroup with very poor prognosis [9]. Conversely, cetuximab and panitumumab combined with chemotherapy can extend median overall survival (OS) by 8 months in half of mCRC patients with RAS wild-type tumors [3]. Differences in chemotherapy response can exist among patients with the same mutation profiles, as patients with KRAS or NRAS wild-type tumors can be partially non-responsive to anti-EGFR agents [10]. Machine learning models combined with mutational status analysis can offer significant advantages in predicting treatment response for colorectal metastases. Therefore, we will examine the mutational profiles of 50 CRC-related genes (e.g., AKT1, ALK, AR, BRAF, etc.) in our mCRC cohort to obtain data used for model building. Chromosomal instability (CIN) is classically defined as an increased rate of numerical or structural chromosomal aberrations in a tumor cell and is present in the vast majority of CRCs (about 85%) [11,12,13]. The presence of CIN can impact prognosis and therapy response, often associated with worse prognosis. Conversely, microsatellite instability (MSI), affecting about 15% of CRC tumors, is considered a good prognostic predictor. Over the past two decades, studies have shown that MSI status is implicated in sensitivity and resistance to adjuvant fluorouracil-based chemotherapy. Therefore, we will perform molecular karyotyping in mCRC patients to examine CIN status. We will also classify CRC tumors based on the number of broad copy number alterations (BCNAs) and identify two MSS tumor subgroups differing in histopathology and gene expression profiles. We will use the first quartile of the BCNA score distribution as a threshold to classify CRC samples as low BCNA (LB) or high BCNA (HB).

#### 2.2.4. Tertiary Objective

The primary goal is to analyze the whole transcriptome in mCRC tumor samples to obtain additional data for constructing learning models. Long non-coding RNAs (lncRNAs) have been implicated in key tumor processes such as epithelial–mesenchymal transition (EMT), immunity, and angiogenesis [5]. Machine learning algorithms have been used to build molecular classifiers based on 16 immuno-related lncRNA signatures (IRLSs) using TCGA and GEO data [14]. These lncRNA signatures can accurately predict tumor response to chemotherapy or targeted therapy in CRC patients, facilitating personalized treatments. We will perform whole-transcriptome analysis in our mCRC cohort to obtain molecular data necessary for model building. The focus will be on analyzing the 16 immuno-related lncRNA signatures (e.g., H19, LINC00308, EpCAM, etc.) for predicting therapy response. The ultimate goal is to evaluate if learning models can be used in clinical trials to assess prognosis or toxicity in standard care settings and as predictors of therapy response.

### 2.3. Experimental Project Objectives

In the retrospective study, up to 150 patients with metastatic colorectal cancer (mCRC) who received chemotherapy alone or in combination with anti-EGFR or anti-VEGF agents between December 2012 and December 2022 will be included. Inclusion criteria are age >18 years; histological diagnosis of mCRC; and assessment of mutational status and microsatellite instability (MSI). The cohort sample size will fall within the range of studies conducted so far [5], and the statistical power will be sufficient to observe statistically significant results at the alpha level of 0.05. Data will be collected retrospectively by reviewing patient medical records at ISMETT. Clinical and pathological diagnosis and therapy data will be collected according to defined standard protocols. Characteristics of commonly used chemotherapy regimens for first-line treatment, such as FOLFIRI, FOLFOX, CapeOX, and XELIRI, will be obtained [15]. Targeted therapy characteristics like anti-EGFR MoAbs, treatments with cetuximab or panitumumab, anti-VEGF MoAbs such as bevacizumab, and immune checkpoint inhibitors (ICIs) will also be available (Xie et al., 2020 [3]). Other characteristics will include cancer information (e.g., T stage, N stage, M stage, microsatellite status, and Karnofsky score), mutational status (K-RAS, N-RAS, BRAF, HER-2 neu), MSI, and demographic variables (e.g., age, gender, and race). Follow-up endpoints such as PFS and OS from diagnosis, first chemotherapy cycle, or primary surgery and mortality data will be appropriately collected until November 2025 to assess prognostic impact.

#### 2.3.1. Biological Sampling

DNA and RNA will be extracted from FFPE tissues using the QIAmp FFPE Tissue kit, RNeasy FFPE kit, and other kits (Qiagen, Zymo Research) according to the manufacturer’s instructions. Samples will be extracted from 10–15 micron FFPE tissue sections with an adequate percentage of neoplastic cells. The quantification of extracted nucleic acids in ng/μL will be performed using Qubit dsDNA and Qubit RNA HS Assay Kits (ThermoFisher Scientific).

#### 2.3.2. Mutation Profiling

The analysis will be performed using Illumina MiSeq and IonS5 platforms (Thermo Fisher). Library preparation will be conducted using a custom pre-designed panel following the manufacturer’s instructions, utilizing a total of 10 ng of input DNA per sample. Template preparation will be manually performed using Ion 520 and Ion 530 Kit-OT2 (Thermo Fisher Scientific) with the Ion OneTouch 2 system or, alternatively, the Ion Chef system if available (Thermo Fisher Scientific). Sequencing will be conducted on the IonS5 semiconductor sequencer using the Ion Torrent SuiteTM browser and Ion ReporterTM version 5.18. Focus will be on hotspot/copy number and gene fusion analysis using the Oncomine Focus Assay panel (Thermo Fisher). Hotspot genes (35 genes) include AKT1, ALK, AR, BRAF, CDK4, CTNNB1, DDR2, EGFR, ERBB2, ERBB3, ERBB4, ESR1, FGFR2, FGFR3, GNA11, GNAQ, HRAS, IDH1, IDH2, JAK1, JAK2, JAK3, KIT, KRAS, MAP2K1, MAP2K2, MET, MTOR, NRAS, PDGFRA, PIK3CA, RAF1, RET, ROS1, and SMO; copy number genes (19 genes) include AKT1, ALK, AR, BRAF, CCND1, CDK4, CDK6, EGFR, ERBB2, FGFR1, FGFR2, FGFR3, FGFR4, KIT, KRAS, MET, MYC, MYCN, PDGFRA, and PIK3CA; and gene fusions (23 genes) include ABL1, AKT3, ALK, AXL, BRAF, EGFR, ERBB2, ERG, ETV1, ETV4, ETV5, FGFR1, FGFR2, FGFR3, MET, NTRK1, NTRK2, NTRK3, PDGFRA, PPARG, RAF1, RET, and ROS1.

#### 2.3.3. Chromosomal Instability

Whole-genome DNA analysis will be performed on Affymetrix arrays (Affymetrix), using 80–500 ng of input DNA depending of the adopted array. SNP 6.0 arrays will assess 906,600 SNPs and 945,826 copy number probes, while Cytoscan arrays will analyze 743,309 SNPs and 1,953,246 copy number probes. OncoScan CNV arrays will be used to analyze DNA from FFPE tumor samples. Array scanning and data analysis will be performed using “GeneChip Command Console” (AGCC) and “Genotyping Console™” (GTC) version 4.2.0 software and Chromosome Analysis Suite (ChAS) 2.1.0 Software or similar.

#### 2.3.4. Whole Transcriptome

Transcriptome analysis will be performed using 0.5–100 ng of total RNA to produce amplified and labeled targets for hybridization with the Human Transcriptome Array 2.0 or Clariom D/S arrays according to Affymetrix protocols. Specifically, both transcriptome arrays contain more than 6.0 million distinct probes to simultaneously analyze at least 44,699 coding and 22,829 non-coding transcripts. Array scanning and data analysis will be performed using Affymetrix Expression and Transcriptome Analysis Consoles, following the manufacturer’s instructions.

#### 2.3.5. lncRNA Analysis

Each lncRNA forming the AI signature will be examined by RT-PCR to further validate the predictive performance of the signature. In brief, total RNA (100 ng) from each sample will be reverse transcribed with the High Capacity cDNA Reverse Transcription Kit (Thermo Fisher); the resulting cDNA will then be used for RT-qPCR with GoTaq Sybr (Promega) on a Rotor-Gene Q (Qiagen, Hilden, Germany). Relative quantification will be performed using LinRegPCR software (11.0), and data will be normalized to glyceraldehyde-3-phosphate dehydrogenase.

#### 2.3.6. Public Databases and Data Availability

The efficacy of therapy response assessment using the learning signature will be investigated in independent public databases, from The Cancer Genome Atlas (TCGA) to Gene Expression Omnibus (GEO), with access to 2277 CRC patients, including those with mCRC. Specifically, therapy databases such as GSE19860, GSE28702, GSE45404, GSE62080, GSE69657, and GSE72970 will be used to evaluate the performance of the learning signature in predicting the benefits of cytotoxic and targeted therapy. Public data that will be used in this work can be acquired from the TCGA Research Network portal (https://portal.gdc.cancer.gov/) and Gene Expression Omnibus (GEO, http://www.ncbi.nlm.nih.gov/geo/).

### 2.4. Study Population

Between 1999 and 2023, a total of 2796 patients underwent surgical procedures at Unit 2. Prior to surgery, each patient underwent a comprehensive review by a multidisciplinary tumor board to evaluate resectability and develop an optimal resection strategy. A total of 1201 therapeutic hepatectomies were conducted at the Department of Hepatobiliary and Transplantation Surgery within Unit 2. Particularly, between 1999 and 2023, a cohort of 486 patients underwent hepatic resection for colorectal liver metastases (CRLM) [14,16]. Of these patients, 428 (88%) underwent open hepatic resection, while 58 (11.9%) underwent laparoscopic liver resection (LLR). The cohort can be stratified into two groups: those who underwent major hepatectomy, defined as the resection of three or more liver segments, and those who underwent minor hepatectomy, involving the resection of fewer than three segments.

In the major hepatectomy group, 85 (17.4%) patients underwent right hepatectomy, with one surgery performed via LLR; 21 patients (4.32%) underwent left hepatectomy; and 3 patients underwent other types of hepatectomy. In the minor hepatectomy group, 30/486 (6.17%) patients underwent left lobectomy, with 5/30 (16.6%) surgeries performed laparoscopically; 23/486 (4.73%) patients underwent bisegmentectomy, with 1/23 (4.54%) surgeries performed using LLR; 34/486 (6.99%) patients underwent segmentectomy, with 3/34 (9.67%) surgeries performed using LLR; 66/486 (13.5%) patients underwent more than three wedge resections, with 8/66 (13.7%) surgeries performed using LLR; 66 (13.5%) patients underwent fewer than three wedge resections, with 6/66 (10%) surgeries performed using LLR; and lastly, single wedge resection was performed in 160 (32.9%) patients, with 34/160 (21.2%) surgeries performed using LLR [17]. The study protocol was submitted to the Ethics Committee and accepted in July 2024. The entry into the Register of Observational Studies occurred at the same time as the protocol submission to the Ethics Committee.

### 2.5. Statistical Analysis

#### 2.5.1. Data Collection Method

Patient characteristics will be retrospectively obtained from ISMETT medical records according to defined standard protocols. Various characteristics of mCRC patients collected from December 2012 to November 2022 will include side (left or right), histology, classification, stage, date of recurrence, PFS, OS, mutational status and MSI, date of death, date of diagnosis, date of first-line therapy, date of primary surgery, chemotherapy, and targeted therapy by reviewing patient medical records, in line with the recent systematic review [5]. Follow-up will be conducted until November 2025. Molecular characteristics will include mutational profiles, whole-transcriptome expression, chromosomal instability (e.g., CIN, BCNA, CNV, and tumor aneuploidy), and individual lncRNAs. FFPE tissues often pose challenges due to degraded DNA/RNA and varying tumor cell content, which can impact the consistency of molecular results. To ensure suitability for molecular results, it is crucial to define inclusion criteria that consider both tumor cell content and nucleic acid quality. Therefore, specify inclusion criteria, e.g., minimum tumor cell content and DNA and RNA quality thresholds, were predefined to ensure sample suitability [18,19]; (1) a minimum of 10% tumor cellularity; (2) a DIN (DNA integrity number) above 7 for DNA; and (3) an RIN (RNA integrity number) above 7 for RNA. To obtain reliable results in molecular assays, such as next-generation sequencing (NGS), and to manage problematic samples, particularly low-quality or low-quantity samples, we will apply targeted strategies [18,19]. These include evaluating DIN, RIN, and fragment size distribution to ensure the proper representation of the target genome and monitoring key metrics such as coverage depth, duplication rates, and read distribution to assess data quality and identify potential issues.

#### 2.5.2. Statistical Power and Primary Objectives

The project will have the statistical power to examine various CRC cancer stages and different treatment options [5]. The primary objectives are to analyze the efficacy of learning algorithms based on molecular data in predicting response to chemotherapy alone or combined with targeted therapy in mCRC patients and to evaluate the efficacy of algorithms in discriminating responders from non-responders. To achieve this, various clinical endpoints, including PFS, OS, and mortality data, will be used to examine therapy response. Clinical and molecular characteristics will be used to construct a database for developing a sophisticated AI-based computational network and validating a predictive learning signature for therapy response. The performance of individual lncRNAs constituting the learning signature will also be examined. Receiver operating characteristic (ROC) analysis and proportional hazards models (Cox regression) will be used to evaluate the predictive performance of models in both training and validation sets. Subsequently, mCRC patients will be classified based on the new AI-derived profiles, and treatments will be chosen based on the best combination of approved therapeutic strategies.

#### 2.5.3. Development of AI-Based Decision Support System

We will develop a platform enabling the risk stratification of patients based on their individual response to treatment with already approved drugs. This will form the basis for developing a decision support system that identifies relevant signatures within patients’ molecular and clinical data. The generalizability of the learning signature will be examined using the TCGA and GEO databases, containing 15,299 lncRNAs and 19,526 protein-coding genes, to exclude overfitting. Moreover, the AI-based decision support system developed in our project will consider sex and gender differences as relevant to the clinical study in accordance with known data or new findings related to the prevalence of CRC in male and female patients (location, hormonal exposure, menopause, etc.). Understanding the mechanism by which sex-related biological factors influence tumor site formation, given that right-sided colon cancer is associated with a worse prognosis and is more common in women than men, is crucial for determining the best screening and treatment protocols, as well as CRC incidence, mortality, and survival rates. This issue will be particularly considered in this study. Combining clinical monitoring with molecular testing will generate new concepts for patients without standard treatment options. Identifying each patient’s individual profile will help choose the correct targeted treatment for mCRC patients. This approach will help overcome the failure of a percentage of patients who do not respond to targeted therapy despite the mutational status of the tumor, as the new combination of treatments, scheduling, and adequate doses will offer new opportunities for mCRC patients. Clinical and molecular profiling has the potential to reveal malignant pathways and identify driver mutations at a specific time point of treatment, offering new opportunities for the targeted use of already established therapeutic interventions in the future. Treatments considered ineffective for an individual patient can be excluded before causing side effects. The timeline for the project is reported in Table 2. Figure 2 reports the patient cohort used to select the number of samples (n.150) for molecular analysis of the test set.

#### 2.5.4. Analytical Plan

We will integrate public datasets (TCGA, GEO) mostly retrieved from the Affymetrix GPL570 platform with a local retrospective mCRC cohort that will also processed using the Affymetrix platform; training datasets will contain similar features to avoid bias or limitations in the comparability of results.

In the retrospective study, the cohort will be divided into a training set and a validation set to train and validate the developed models, avoiding overfitting. The training set will be used to set up learning models using n-fold cross-validation to estimate meta-parameters. Techniques such as plotting learning curves will be also used to detect overfitting and underfitting. Outliers in training data can inflate model noise or reduce generalizability; therefore, such outliers will be identified and removed from the dataset. When applying machine learning methods, quality of the dataset is fundamental, and a preprocessing strategy will be employed for improving data quality and fit in ML methods, such as dimensionality reduction, feature selection, and feature extraction [20]. Dimensionality reduction by eliminating irrelevant data together with the creation of new features that are a subset of old ones and contain all the relevant information can reduce noise and produce performant learning models.

Tests will be used to compare the performance of different models when necessary. To select candidate biomarkers from molecular data, a penalized Cox regression approach with cross-validation will be applied, with PFS or OS as outcomes. Subsequently, we will integrate various machine learning algorithms, such as random survival forests, Lasso, support vector machines, decision trees, and artificial neural networks, to develop a predictive learning signature with high precision and stability. Combinations of algorithms will be applied to the signatures of prognostic molecular biomarkers to fit prediction models; for each model, performance will be calculated across the validation dataset, and the model with the highest predictive accuracy will be considered optimal. In the case that performance remains suboptimal, techniques such as data augmentation and synthetic data generation can potentially be employed to enhance the training process [21]. Outcome case distribution will be considered in training and test sets [14]. Model performance will be evaluated using the area under the ROC curve or proportional hazards models (Cox regression) and estimates of sensitivity (SE) and specificity (SP), and their 95% CIs will be reported. SE will refer to the test’s ability to correctly identify patients who are responders, while SP estimates the ability to correctly classify those who are non-responders [22,23,24]. Stratifying responding patients based on the AI learning signature will enable the design of a personalized therapeutic approach. Finally, the inclusion of an external validation cohort with truly independent data, beyond random splits or reusing public archives, will be considered. In Figure 3, the workflow of the protocol is shown.

## 3. Results and Discussion

The relevance of this study and its contribution to the literature are based on the first use of AI networks to build a signature predictive of the response to chemotherapy in mCRC patients using transcriptomic and genomic data in real-world conditions. The novelty is in the attempt to answer to the basic question of whether an AI-driven signature can guide clinicians to choose chemotherapies for cancer patients on an individualized basis by identifying responders and non-responders.

Our learning model based on molecular data will be able to identify new treatment sequences with available and approved molecularly targeted drugs for mCRC patients at a higher level compared to conventional diagnostic and therapeutic methods (Mullard, 2020). A new classification of mCRC based on learning parameters will contribute to improved clinical management of mCRC and clinical decision-making. Clinical response learning models will be used to predict individual patient response outcomes and stratify mCRC patients based on the best combination of regulatory-approved therapeutic strategies, as well as to develop personalized treatment decision processes for mCRC patients with poor performance status and low survival outcomes [25]. The project will help identify personalized treatments with the highest probability of success to improve patient outcomes, reduce costs, and enhance healthcare efficiency. We have demonstrated that all AI investigations analyzing complex data conducted in this field have generally provided promising results in predicting therapy response using a meta-analytical approach [5]. The sample size should be sufficient to achieve the proposed objectives, in line with investigations conducted so far, and we should have sufficient data to populate the AI models. The follow-up should also be sufficient, as we will look back in time and use pre-existing clinical and pathological data to examine predictive models of therapy response. Moreover, the use of an external validation arm, instead random splits or reusing public archives, will be evaluated. Moreover, there is extensive involvement of oncologists, and our research group has experience in this subject, as we participated in the EU-funded study titled “Targeted therapy for patients with advanced colorectal cancer, REVERT” (https://www.revert-project.eu/). Therefore, the risk involved in this project is low, while the potential upside is high, and the learning signature will provide useful information to make therapeutic decisions more quickly.

Few therapeutic options are available for resistant mCRC, which might be challenging to address through non-cross-resistant anti-tumor drugs for second and subsequent lines of therapeutic methods [2]. Adaptive resistance can occur at any time during treatment, causing therapeutic failures and subsequently high mortality rates [3]. Regarding the translational scope that our study might have in the future, our predictive models will be useful in selecting the most effective and cost-efficient therapeutic interventions for cancer patients. We will identify the machine learning molecular profiles of mCRC patients who can be included in stratification models for predictive treatment response and who can potentially benefit from new combinatorial therapy. Patients will be classified based on the new resulting AI profiles, and treatment will be chosen based on the best combination of approved therapeutic strategies.

Although mCRC, like many other cancers, is affected by a significant degree of spatial and temporal heterogeneity [26], the selected panel of biomarkers will allow us to assess the mutational profile of genes involved in the main mechanisms of primary/secondary resistance [5,27]. Furthermore, the result of the mutational profile analysis obtained from this panel will allow patients access to the drugs available in current clinical practice as well as to the main clinical trials designed for this disease setting, thus meeting real-world needs.

There are many potential benefits of the use AI technologies techniques in the field of cancer research; nevertheless, it is critical to report some potential limitations of this kind of study protocol. One area of concern is the high technical complexity of learning networks, which needs to be addressed to move toward therapeutic interventions. The lack of explainability criteria and understanding how mathematical models reach their conclusions could be a great barrier to overcome before implementing the use of machine learning model-based devices in clinical practice [28]. Also, merely combining existing public data with a local cohort can still lead to an overrepresentation of certain subpopulations, undermining generalizability.

Advances in AI and machine learning have revolutionized colorectal cancer (CRC) research, driving innovations that span histopathological image analysis, multi-omics integration, and predictive modeling. For instance, the hybrid model “Deep Feature-Based Broad Learning System” (DeepBLS) has significantly enhanced tissue phenotyping from whole-slide images, providing detailed insights into CRC histopathology [29]. Additionally, the integration of visual and semantic features through artificial intelligence-based approaches has refined tissue characterization, while convolution transformer-based networks with adaptive convolution and dynamic attention mechanisms have improved image classification accuracy [30,31]. Complementary approaches integrate multi-omics data to inform immunotherapy and lead to the development of immune-derived lncRNA prognostic signatures [14,32]. Moreover, lipidomic biomarkers analyzed via machine learning have facilitated more precise staging of CRC [33]. Regression-based deep learning frameworks have enabled the prediction of molecular biomarkers directly from histopathological data, aiding in the identification of metastatic indicators and enhancing gene-mutation algorithms for forecasting therapeutic responses [34,35,36]. Moreover, integrated platforms also support prognosis prediction and drug response modeling in hepatic metastases, while genetic algorithm-enhanced artificial neural networks provide non-invasive assessments of microsatellite instability [37,38]. Machine learning methodologies have further elucidated the tumor microenvironment and identified microbial and genomic signatures that distinguish right- from left-sided CRC [39,40]. Multistain deep learning techniques have improved accuracy in therapy response and prognosis, and the combination of convolutional neural networks and support vector machines has further shown promise in predicting CRC prognosis and mutational signatures from standard hematoxylin and eosin staining images [41,42]. Concurrently, integrative bioinformatics pipelines have uncovered novel diagnostic gene biomarkers, while machine learning-driven frameworks aid in the diagnosis and management of CRC liver metastases, stratify lung metastasis risk in highly imbalanced datasets, and screen for liver metastasis-specific genes [43,44,45,46]. Collectively, these multifaceted advancements accentuate the pivotal role of machine learning in refining CRC diagnostics, prognostics, and therapeutic strategies, ultimately advancing personalized medicine.

Recently, immune checkpoint inhibitors, acting by blocking immunoinhibitory signals and enabling patients to induce an anti-tumor action, were shown to be active in the treatment of mCRC [47]. Immune checkpoint inhibitors targeting PD-1 were demonstrated to be effective in a selected mCRC patient category having tumors displaying dMMR or MSI-H [48]. Also, the neutrophil-to-lymphocyte ratio (NLR) can be used to reflect the body’s inflammatory status, with prognostic value in different cancers. In particular, NLR is used as a biomarker in CRC liver metastases, showing a significant association with poor oncological outcomes. A recent systematic review analyzed the NLR ratio as a prognostic factor in mCRC patients treated with various local therapies [49]. In the article, the influence of preoperative NLR on the prognosis of mCRC patients receiving surgery was analyzed using a meta-analytic approach considering overall survival (OS), disease-free survival (DFS), and relapse-free survival (RFS) as the main indicators of response to therapy. The meta-analysis showed that high preoperative NLR was associated with poor OS [hazard ratio (HR) of 1.83, 95% C.I. 1.61–2.08, *p* < 0.01], DFS (HR of 1.78, 95% C.I. 1.16–2.71, *p* < 0.01), and RFS (HR of 1.46, 95% C.I. 1.15–1.85, *p* < 0.01). Conversely, it was shown that lymphocyte-to-monocyte ratio (LMR) predicted survival after radiofrequency ablation (RFA) for colorectal liver metastasis [50]. OS was significantly higher in mCRC patients with LMR > 3.96% [55 months (95% C.I., 37–69)] than in those with LMR ≤ 3.96% [34 months (95%CI, 26–39)]. The response to anti-epidermal growth factor receptor (EGFR) therapy with or without irinotecan regimens has been examined in mCRC patients with wild-type KRAS by a machine learning approach using immunohistochemical data [51]. In the study, the percentage of positively stained tumor cells within the tumor areas for the EGFR ligands amphiregulin (AREG) and epiregulin (EREG) were demonstrated to predict benefit from the anti-EGFR agent panitumumab. Increased PFS was linked to the response to EGFR therapy in patients carrying wild-type KRAS and high ligand expression with respect to those having standard regimens (8.0 vs. 3.2 months, HR of 0.54, 95% C.I. 0.37–0.79) and in those carrying both wild-type KRAS and BRAF, with an HR of 0.53 (0.36–0.78). Different machine learning networks have been used to build and validate an immune-related signature capable of predicting the response to systematic therapy with or without anti-vascular endothelial growth factor (VEGF) therapy [14]. In this model, 235 lncRNA modulators of immune-related pathways, including the T-cell receptor signaling and antigen processing and presentation pathways, were examined by integrated networks using molecular data derived from immune infiltration patterns of the public TCGA-CRC dataset. The final signature formed by 16 predictive lncRNAs was shown to predict the response to fluorouracil-based regimens for metastatic or recurrent CRC, as well as indicating the benefits of adding bevacizumab.

Table 3 reports the biomarkers that have been used to develop and validate predictive models of response to therapy in mCRC [14,33,36,40,43,51,52,53,54,55,56,57,58,59,60,61]. Most of the studies have benefited from the availability of high-throughput information from different genomic characterization levels, such as genome, transcriptome, proteome, and metabolome. Table 3 shows that microarray gene expression data are more often chosen for building molecular biomarkers based classifiers [40,43,52,54,57,58]. Networks built using NGS and genotyping data are also used as inputs for AI learning models aimed at classifying mCRC patients into responders vs. non-responders [36,56,59,60,61].

## 4. Conclusions

Today, it is well known that every mCRC patient requires individual specific therapy. Furthermore, when a genetic variant subline is resistant to treatment, few treatment options are available for resistant mCRC, which can be difficult to tackle through non-cross-resistant anticancer drugs for second and subsequent lines of therapy despite new diagnostic and therapeutic methods. Therefore, more research is needed for the identification of comprehensive clinical and molecular signatures predictive of efficacy before mCRC patients can be randomized for treatment based on a predictive model vs. usual treatment. The proposed platform could significantly improve personalized therapy for mCRC patients by optimizing therapeutic choices, reducing costs, and enhancing healthcare efficiency.

## Figures and Tables

**Figure 1 life-15-00320-f001:**
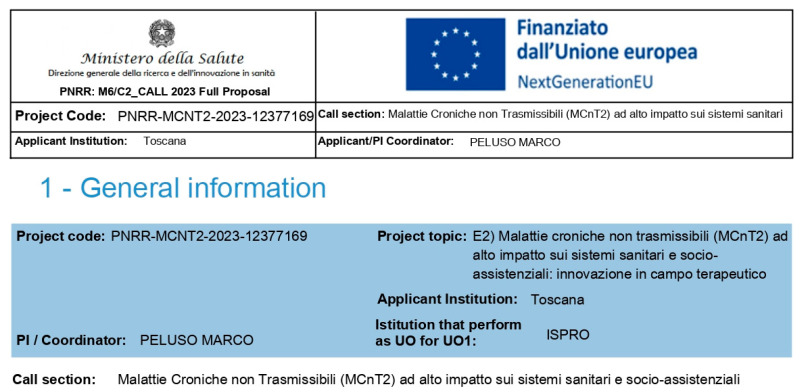
General information and financing institutions.

**Figure 2 life-15-00320-f002:**
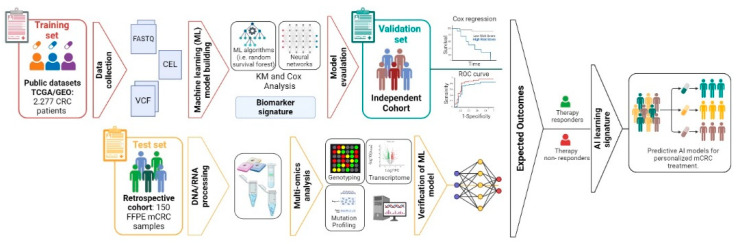
Patient population used to select retrospective cohort.

**Figure 3 life-15-00320-f003:**
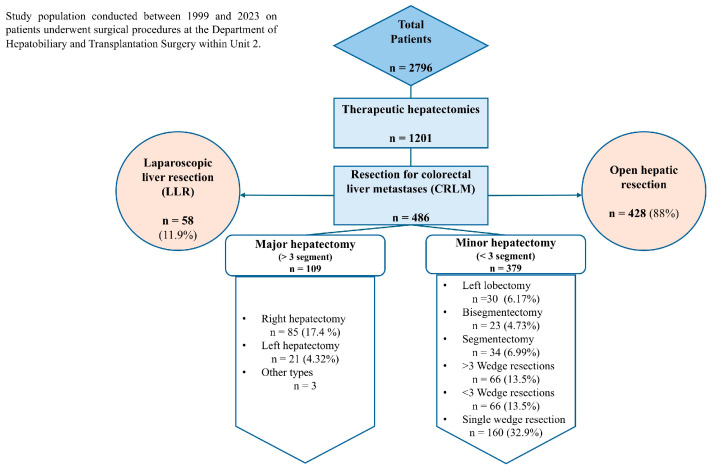
Protocol workflow.

**Table 1 life-15-00320-t001:** Description of the activities of each unit throughout the project.

Institution Department/Division/Laboratory	Operational Unit	Activities
Research and Development Branch, Regional Cancer Prevention Laboratory, Cancer Prevention Laboratory, ISPRO-Study, Prevention and Oncology Network Institute.	Unit 1	Responsible for the epidemiological plan. Overseeing clinical, laboratory, statistical, and bioinformatic activities. Validating and developing the supervised learning algorithm based on molecular biomarker data and identifying the newly constructed molecular classifier. Validating overall transcription results obtained by testing selected long non-coding RNAs via RT-PCR.
Department for the Treatment and Study of Abdominal Diseases and Abdominal Transplantation. Istituto di Ricovero e Cura a Carattere Scientifico-Istituto Mediterraneo per i Trapianti e Terapie ad Alta Specializzazione (IRCCS-ISMETT). University of Pittsburgh Medical Center (UPMC).	Unit 2	Collecting archived FFPE tumor samples (primary and metastatic lesions) from patients who underwent surgery over the past 10 years and received standard chemotherapy with or without anti-EGFR or anti-VEGF agents. Collecting clinical and pathological diagnosis data according to defined standard protocols.
Department of Surgical, Oncological and Oral Sciences, University of Palermo.	Unit 3	Extracting and purifying RNA and DNA from FFPE samples. Analyzing mutational profiles of 50 CRC-related genes. Examine copy number variants for a series of relevant genes.
Department of Biomedical and Biotechnological Sciences, University of Catania. Department of General Surgery and Medical-Surgical Specialties, University of Catania.	Unit 4	Performing whole-transcriptome and molecular karyotype analyses to analyze IncRNA signatures. Conducting whole-genome DNA analysis. Examining chromosomal instability to measure chromosomal aberration rates in tumors and metastases of CRC patients. Identifying patient subgroups with different prognoses and therapy responses utilizing a CRC tumor classification based on the BCNA number. Validating overall transcription results obtained by testing selected long non-coding RNAs via RT-PCR.

**Table 2 life-15-00320-t002:** Study protocol timeline.

Tasks	Timeline	Note
Study duration	2 years	6-month extension upon request
Ethics Committee approval	1 month	Obtained before grant agreement
Retrospective study timeline	10 years	
Collection, purification, and shipment of biological samples	20 months	
RNA and DNA analysis using Affymetrix	20 months	
Mutation analysis using NGS	20 months	
lncRNA signature analysis using RT-qPCR	17 months	
Follow-up for endpoints ^a^ and mortality	Up to six or more months ^b^	
Statistical and bioinformatic analyses	19 months	

^a^ PFS, OS, TTNT, or RECIST criteria endpoints; ^b^ From the last cycle of chemotherapy for the entire project duration.

**Table 3 life-15-00320-t003:** Main biomarkers used in the field of artificial intelligence predictive models of response to therapy in metastatic colorectal cancer patients.

Techniques	Biomarkers	Reference
Microarray gene expression data	74 genes	[57]
Microarray gene expression data	4 genes	[46]
Microarray differentially expressed (DE) gene profiles	18 genes	[58]
Microarray DE gene profiles	14 genes	[54]
Microarray DE gene profiles	14 genes	[52]
Microarray DE gene profiles	9 genes	[43]
Illumina HiSeq (DE) gene profiles	5, 46, and 54 genes	[40]
Next-generation sequencing (NGS) data	67 genes	[56]
NGS data	7 genes	[36]
Genome-wide genotyping data	781 SNPs	[60]
Genotyping data	27 SNPs	[59]
Exome sequencing data	1 and 2 SNPs	[61]
Cytokine expression	17 cytokines	[55]
Cytokine expression	Lipid biomarkers and 3 cytokines	[33]
Immune infiltration data	16 long non-coding RNAs	[14]
Immunochemistry (IHC) data	Amphiregulin/epiregulin	[51]
Protein patterns	6 and 7 proteins	[53]

## Data Availability

The data supporting reported results can be obtained upon reasonable requested.

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
