# Peer review of "Clinical Validation of a Machine Learning-Based Biomarker Signature to Predict Response to Cytotoxic Chemotherapy Alone or Combined with Targeted Therapy in Metastatic Colorectal Cancer Patients: A Study Protocol and Review"

_life, 2025, doi:10.3390/life15020320_

Round 1

Reviewer 1 Report

Comments and Suggestions for Authors

Dear Authors, 

Comment 1:

I recommend organizing the paper into distinct sections for better clarity and readability. For example:

Introduction

Materials and Methods

Results

Discussion

Conclusions

Comment 2:

At the end of the Introduction section, please explicitly list your key contributions in this work to clearly highlight the novelty and importance of your study.

Comment 3:

Consider adding a dedicated Literature Review section. Include recent studies from 2023 and 2024 to provide a comprehensive overview of related work. For instance:

"Artificial intelligence-based tissue phenotyping in colorectal cancer histopathology using visual and semantic features aggregation."

"A novel convolution transformer-based network for histopathology-image classification using adaptive convolution and dynamic attention."

Comment 4:

The implementation details appear to be missing. Please include a description of the experimental setup, model training parameters, and how issues like underfitting and overfitting were addressed during the training process.

Comment 5:

It would improve the clarity of the results section if the findings are presented in a tabular format for better interpretation and comparison.

Author Response

Thank you for your valuable review, please see my reply attached.

Reviewer 2 Report

Comments and Suggestions for Authors

Scope and Innovation

  • The study addresses a relevant clinical need—predicting therapy response in metastatic colorectal cancer (mCRC) using molecular data and AI. However, it remains unclear whether the chosen biomarkers (e.g., MSI, mutational profiles, etc.) will provide sufficient predictive power in the real-world setting, given the known heterogeneity of mCRC.

Methodological Rigor

  • The plan to integrate public datasets (TCGA, GEO) with a local retrospective cohort is commendable, but the protocol does not thoroughly describe how discrepancies between different data sources (e.g., variations in sample processing, data quality) will be handled. This could introduce bias or limit the comparability of results.
  • The model-building section outlines diverse machine learning methods (Lasso, neural networks, etc.), but lacks detail on how final algorithms will be selected or combined. The risk of overfitting is mentioned, yet no clear strategy is proposed if performance remains suboptimal.
  • While cross-validation is included, there is no mention of external validation with truly independent data—beyond random splits or reusing public archives—nor a plan for prospective validation, which may reduce confidence in the clinical application of the proposed models.

Data Collection and Quality Control

  • FFPE tissue quality and the potential for degraded DNA/RNA could undermine the consistency of molecular results. The manuscript acknowledges extraction steps but does not specify predefined inclusion criteria (e.g., minimum tumor cell content, quality thresholds) to ensure sample suitability.
  • Detailed plans on how outlier or problematic samples will be managed are absent. Failure to address these issues up front could inflate model noise or reduce generalizability.

Ethical and Practical Considerations

  • The ethical statement confirms IRB approval, but the limited timeline to gather retrospective data and perform full multi-omics analysis raises concerns about whether all steps (sample retrieval, analysis, data cleaning) can be feasibly completed within two years.
  • Plans for re-consent or data privacy (especially for older archival samples) are not thoroughly described, which may lead to logistical or compliance hurdles.

Potential Impact

  • The proposed platform could significantly improve personalized therapy if successfully validated, but the lack of a robust external validation phase is a notable gap. Merely combining existing public data with a local cohort may still lead to an overrepresentation of certain subpopulations, undermining the “generalizability” claim.
  • Relying heavily on molecular biomarkers for AI models is ambitious. Without a clear fallback plan should certain biomarkers or lncRNAs prove uninformative, the entire pipeline risks producing inconclusive outcomes.

Overall Assessment

  • While the study protocol is conceptually strong and addresses an unmet clinical need, a clearer, more rigorous plan for external validation and data quality control is needed to ensure that the model’s results will truly be applicable in diverse clinical contexts. Addressing these concerns prior to implementation would substantially strengthen the protocol’s reliability and eventual impact.

Author Response

(The authors gave the same response as above.)

Reviewer 3 Report

Comments and Suggestions for Authors

AI would probably play an important role in thr medicine, but there is still a lot of work.

Author Response

(The authors gave the same response as above.)

Reviewer 4 Report

Comments and Suggestions for Authors

The authors present a well-organized, well written, and thorough research study protocol. While there are no results to discuss or report at this time, there are a few minor queries that must be addressed:

1. The authors must emphasize the relevance of their study and contribution to the literature. In other words, if this has been done before, how does this study contribute anything new, relevant, or important? If not, and if novel, they must state it so.

2. Graphical adjuncts such as figures, diagrams, or tables would enhance the manuscript and are highly recommended for our visual readers.

3. In the discussion section, we need a few sentences of weaknesses and limitations associated with the methodology chosen for this study protocol.

I look forward to revising the protocol after these minor edits.

Author Response

(The authors gave the same response as above.)

Reviewer 5 Report

Comments and Suggestions for Authors

The protocol is well written and clear. Since it is a protocol, so without study data or results, the authors should describe better the background underlying the rationale to the study, for example the authors could comment on the immune-related biomarkers in CRC liver metastases (for exampkle cite the series PMID: 27122671).

A table summarizing the main biomarkers in this field would improve the quality of the paper

The authors should make explicit in the title which kind of therapy is considered (only systemic therapy? Surgery?)

I think the references were not reported according to the journal's requirements

Author Response

(The authors gave the same response as above.)

Round 2

Reviewer 4 Report

Comments and Suggestions for Authors

The researchers have improved the manuscript and according to the queries raised by the reviewers. I have carefully examined their answers and agree with the manuscript and its current status. I approve for publication of the editor in chief agrees. Congratulations to the authors.

Author Response

Dear Editor,

We prepared the final version of the manuscript entitled Clinical validation of a machine learning-based biomarkers signature to predict response to therapy in metastatic colorectal cancer patients: a study protocol and a review by Duilio Pagano, Vincenza Barresi, Alessandro Tropea, Antonio Galvano, Viviana Bazan, Adele Caldarella, Cristina Sani, Gianpaolo Pompeo, Valentina Russo, Rosa Liotta, Chiara Scuderi, Simona Mercorillo, Floriana Barbera, Noemi Di Lorenzo, Agita Jukna, Valentina Carradori, Romina D’Aurizio, Monica Rizzo, Salvatore Gruttadauria, Marco Peluso according to the requests of the reviewer N.5, we addressed the following answers as requested. We hope that the manuscript complies with “Life” guidelines and everything is in order for acceptance and publication.

Sincerely

Answer to the Questions of the Reviewer N.5

Question N.1 Since it is a protocol, so without study data or results, the authors should describe better the background underlying the rationale to the study, for example the authors could comment on the immune-related biomarkers in CRC liver metastases (for exampkle cite the series PMID: 27122671).

Answer: We agree with the reviewer and the following paragraph, where we cited some papers including PMID: 27122671, was added to the Results and Discussion section at lines 533-568: Recently, immune checkpoint inhibitors, acting by blocking immunoinhibitory signals and enabling patients to induce an anti-tumor action, were shown to be active in the treatment of mCRC [1]. Immune checkpoint inhibitors targeting PD-1 demostrated to be effective in a selected mCRC patient category having tumors displaying dMMR or MSI-H [2]. Also, the neutrophil-to-lymphocyte ratio (NLR) can be used to reflect body's inflammatory status with prognostic value in different cancers. In particular, NLR is used as biomarker in in CRC liver metastases showing significant association with poor oncological outcomes. A recent systematic review analyzed the NLR ratio as a prognostic factor in mCRC patients treated with various local therapies [3]. In that article, the influence of preoperative NLR in the prognosis of mCRC patients receiving surgery was analyzed using a meta-analytic approach and considering the overall survival (OS), the disease-free survival (DFS) and the relapse-free survival (RFS) as the main indicators of response to therapy. The meta-analysis showed that high preoperative NLR was associated  with poor OS [hazard ratio (HR) of 1.83, 95% C.I. 1.61–2.08, p<0.01], DFS (HR of 1.78, 95% C.I. 1.16–2.71, p<0.01) and RFS (HR of 1.46, 95% C.I. 1.15–1.85, p<0.01). Conversely, it was shown that lymphocyte-to-monocyte ratio (LMR), predicted survival after after radiofrequency ablation (RFA) for colorectal liver metastasis  [4].  OS was significantly increased in mCRC patients with LMR of > 3.96% [55 months (95% C.I., 37-69)] than in those with LMR ≤ 3.96% [34 months (95%CI, 26-39)]. The response to anti-epidermal growth factor receptor (EGFR) therapy with or without irinotecan regimens has been examined in mCRC patients with KRAS wild-type even by a machine learning approach using immunohistochemical data [5]. In that study, the percentage of positively stained tumor cells within the tumor areas for EGFR ligands amphiregulin (AREG) and epiregulin (EREG) were demonstrated to predict benefit from the anti-EGFR agent panitumumab. Increased PFS was linked to the response to EGFR therapy in patients carrying KRAS wild-type and high ligand expression in respect to those having standard regimens (8.0 vs. 3.2 months, HR of 0.54, 95% C.I. 0.37–0.79), and in those carrying both KRAS and BRAF wild-type, HR of 0.53 (0.36–0.78). Different machine learning networks have been even used to build and validate an immune related signature capable to predict the response to systematic therapy with or without anti-vascular endothelial growth factor (VEGF) therapy [6]. In this model, 235 lncRNA modulators of immune-related pathways, including the T cell receptor signaling and antigen processing and presentation pathways, were examined by integrated networks using molecular data derived from immune infiltration patterns of the public TCGA-CRC dataset. The final signature formed by 16 predictive lncRNAs was shown to predict the response to fluorouracil-based regimens for metastatic or recurrent CRC patients, as well as the benefits of adding bevacizumab.

Question N.2 A table summarizing the main biomarkers in this field would improve the quality of the paper

Answer: A table summarizing the main biomarkers in AI field was added to the Results and Discussion section at lines 570-581: Table 3 reports the biomarkers that have been used to develop and validate predictive models of response to therapy in mCRC [5-20]. Most of the studies have benefited from the availability of high-throughput information from different genomic characterization levels, as genome, transcriptome, proteome, and metabolome. Table 3 shows that microarray gene expression data were more often chosen for building molecular biomarkers based classifiers [7,9,12,13,15,16]. Networks build using NGS and genotyping data were also used to fed as inputs learning AI models aimed to classify mCRC patients in responders vs. non-responders [11,14,17-19].

Table 3. Main biomarkers used in the field of artificial intelligence predictive models of response to therapy in metastatic colorectal cancer patients.

Tecniques

Biomarkers

Reference

Microarray gene-expression data

74-genes

[12]

Microarray gene-expression data

4-genes

[21]

Microarray differentially expressed (DE) gene profiles

18-genes

[13]

Microarray DE gene profiles

14-genes

[9]

Microarray DE gene profiles

14-genes

[7]

Microarray DE gene profiles

9-genes

[16]

Illumina HiSeq (DE) gene profiles

5-, 46-, 54-genes

[15]

Next generation sequencing (NGS) data

67-genes

[11]

NGS data

7-genes

[18]

Genome wide genotyping data

781-SNPs

[17]

Genotyping data

27 SNPs

[14]

Exome-sequencing data

1, 2-SNPs

[19]

Cytokine expression

17-cytokines

[10]

Cytokine expression

lipid biomarkers and 3-cytokines

[20]

Immune infiltration data

16-long noncoding RNA

[6]

Immunochemistry (IHC) data

Amphiregulin/epiregulin

[5]

Protein patterns

6-, 7-proteins

[8]

Question N.3 The authors should make explicit in the title which kind of therapy is considered (only systemic therapy? Surgery?)

Answer: We made explicit in the title which kind of therapy is considered: Clinical validation of a machine learning-based biomarkers signature to predict response to cytotoxic chemotherapy alone or combined to targeted therapy in metastatic colorectal cancer patients: a study protocol and a review

Question N.4 I think the references were not reported according to the journal's requirements

Answer: references are now reported according to the MDPI's requirements

References

  1. Mahoney, K.M.; Rennert, P.D.; Freeman, G.J. Combination cancer immunotherapy and new immunomodulatory targets. Nature Reviews Drug Discovery 2015, 14, 561-584, doi:10.1038/nrd4591.
  2. Dai, Y.; Zhao, W.; Yue, L.; Dai, X.; Rong, D.; Wu, F.; Gu, J.; Qian, X. Perspectives on Immunotherapy of Metastatic Colorectal Cancer. Frontiers in Oncology 2021, 11.
  3. Li, Y.; Xu, T.; Wang, X.; Jia, X.; Ren, M.; Wang, X. The prognostic utility of preoperative neutrophil-to-lymphocyte ratio (NLR) in patients with colorectal liver metastasis: a systematic review and meta-analysis. Cancer Cell International 2023, 23, doi:https://doi.org/10.1186/s12935-023-02876-z.
  4. Facciorusso, A.; Del Prete, V.; Crucinio, N.; Serviddio, G.; Vendemiale, G.; Muscatiello, N. Lymphocyte-to-monocyte ratio predicts survival after radiofrequency ablation for colorectal liver metastases. World journal of gastroenterology 2016, 22, 4211-4218, doi:DOI: 10.3748/wjg.v22.i16.4211.
  5. Williams, C.; Seligmann, J.F.; Guetter, C.; Zhang, L.; Yan, D.; Muranyi, A.; Bai, I.; Singh, S.; Elliott, F.; Shires, M.; et al. Artificial intelligence-assisted immunohistochemical (IHC) evaluation of tumor amphiregulin (AREG) and epiregulin (EREG) expression as a combined predictive biomarker for panitumumab (Pan) therapy benefit in RAS wild-type (wt) metastatic colorectal cancer (mCRC): Analysis within the phase III PICCOLO trial. Journal of Clinical Oncology 2021, 39, 111-111, doi:10.1200/JCO.2021.39.3_suppl.111.
  6. Liu, Z.; Liu, L.; Weng, S.; Guo, C.; Dang, Q.; Xu, H.; Wang, L.; Lu, T.; Zhang, Y.; Sun, Z.; et al. Machine learning-based integration develops an immune-derived lncRNA signature for improving outcomes in colorectal cancer. Nature Communications 2022, 13, 816, doi:10.1038/s41467-022-28421-6.
  7. Del Rio, M.; Molina, F.; Bascoul-Mollevi, C.; Copois, V.; Bibeau, F.d.r.; Chalbos, P.; Bareil, C.; Kramar, A.; Salvetat, N.; Fraslon, C.; et al. Gene expression signature in advanced colorectal cancer patients select drugs and response for the use of leucovorin, fluorouracil, and irinotecan. Journal of clinical oncology : official journal of the American Society of Clinical Oncology 2007, 25, 773-780, doi:10.1200/jco.2006.07.4187.
  8. Yuan, Y.; Shen, H.; Li, M.-D.; Tan, C.-W.; Yu, J.-K.; Fang, X.-F.; Zheng, S. Identification of the biomarkers for the prediction of efficacy in the first-line chemotherapy of metastatic colorectal cancer patients using SELDI-TOF-MS and artificial neural networks. Journal of Clinical Oncology 2012, 30, e14026-e14026, doi:10.1200/jco.2012.30.15_suppl.e14026.
  9. Tsuji, S.; Midorikawa, Y.; Takahashi, T.; Yagi, K.; Takayama, T.; Yoshida, K.; Sugiyama, Y.; Aburatani, H. Potential responders to FOLFOX therapy for colorectal cancer by Random Forests analysis. British journal of cancer 2012, 106, 126-132, doi:10.1038/bjc.2011.505.
  10. Chen, Z.-Y.; He, W.-Z.; Peng, L.-X.; Jia, W.-H.; Guo, R.-P.; Xia, L.-P.; Qian, C.-N. A prognostic classifier consisting of 17 circulating cytokines is a novel predictor of overall survival for metastatic colorectal cancer patients. INTERNATIONAL JOURNAL OF CANCER 2014, 136, 584-592, doi:https://doi.org/10.1002/ijc.29017.
  11. Abraham, J.P.; Magee, D.; Cremolini, C.; Antoniotti, C.; Halbert, D.D.; Xiu, J.; Stafford, P.; Berry, D.A.; Oberley, M.J.; Shields, A.F.; et al. Clinical Validation of a Machine-learning derived Signature Predictive of Outcomes from First-line Oxaliplatin-based Chemotherapy in Advanced Colorectal Cancer. Clinical Cancer Research 2021, 27, 1174-1183, doi:10.1158/1078-0432.ccr-20-3286.
  12. Tian, S.; Wang, F.; Lu, S.; Chen, G. Identification of Two Subgroups of FOLFOX Resistance Patterns and Prediction of FOLFOX Response in Colorectal Cancer Patients. Cancer investigation 2021, 39, 62-72, doi:10.1080/07357907.2020.1843662.
  13. Lu, W.; Fu, D.; Kong, X.; Huang, Z.; Hwang, M.; Zhu, Y.; Chen, L.; Jiang, K.; Li, X.; Wu, Y.; et al. FOLFOX treatment response prediction in metastatic or recurrent colorectal cancer patients via machine learning algorithms. Cancer medicine 2020, 9, 1419-1429, doi:https://doi.org/10.1002/cam4.2786.
  14. Naseem, M.; Cao, S.; Yang, D.; Millstein, J.; Puccini, A.; Loupakis, F.; Stintzing, S.; Cremolini, C.; Tokunaga, R.; Battaglin, F.; et al. Random survival forests identify pathways with polymorphisms predictive of survival in KRAS mutant and KRAS wild-type metastatic colorectal cancer patients. Scientific Reports 2021, 11, 12191, doi:10.1038/s41598-021-91330-z.
  15. Kolisnik, T.; Sulit, A.K.; Schmeier, S.; Frizelle, F.; Purcell, R.; Smith, A.; Silander, O. Identifying important microbial and genomic biomarkers for differentiating right- versus left-sided colorectal cancer using random forest models. BMC cancer 2023, 23, 647, doi:10.1186/s12885-023-10848-9.
  16. Vaziri-Moghadam, A.; Foroughmand-Araabi, M.-H. Integrating machine learning and bioinformatics approaches for identifying novel diagnostic gene biomarkers in colorectal cancer. Scientific Reports 2024, 14, 24786, doi:10.1038/s41598-024-75438-6.
  17. Ubels, J.; Schaefers, T.; Punt, C.; Guchelaar, H.-J.; de Ridder, J. RAINFOREST: a random forest approach to predict treatment benefit in data from (failed) clinical drug trials. Bioinformatics 2020, 36, i601-i609, doi:10.1093/bioinformatics/btaa799.
  18. Johnson, H.; El-Schich, Z.; Ali, A.; Zhang, X.; Simoulis, A.; Wingren, A.G.; Persson, J.L. Gene-Mutation-Based Algorithm for Prediction of Treatment Response in Colorectal Cancer Patients. Cancers 2022, 14, 2045.
  19. Barat, A.; Smeets, D.; Moran, B.; Zhang, W.; Cao, S.; Das, S.; Klinger, R.; Betge, J.; Murphy, V.; Bacon, O.; et al. Combination of variations in inflammation- and endoplasmic reticulum-associated genes as putative biomarker for bevacizumab response in KRAS wild-type colorectal cancer. Scientific Reports 2020, 10, 9778, doi:10.1038/s41598-020-65869-2.
  20. Krishnan, S.T.; Winkler, D.; Creek, D.; Anderson, D.; Kirana, C.; Maddern, G.J.; Fenix, K.; Hauben, E.; Rudd, D.; Voelcker, N.H. Staging of colorectal cancer using lipid biomarkers and machine learning. Metabolomics 2023, 19, 84, doi:10.1007/s11306-023-02049-z.
  21. Zheng, S.; He, H.; Zheng, J.; Zhu, X.; Lin, N.; Wu, Q.; Wei, E.; Weng, C.; Chen, S.; Huang, X.; et al. Machine learning-based screening and validation of liver metastasis-specific genes in colorectal cancer. Scientific Reports 2024, 14, 17679, doi:10.1038/s41598-024-68706-y.

Reviewer 5 Report

Comments and Suggestions for Authors

The authors did not address any of my points. I even wonder if they read my comments.....

Author Response

(The authors gave the same response as above.)

Round 3

Reviewer 5 Report

Comments and Suggestions for Authors

The manuscript is OK